# Evaluation of Two Species of Macroalgae from Azores Sea as Potential Reducers of Ruminal Methane Production: In Vitro Ruminal Assay

**DOI:** 10.3390/ani14060967

**Published:** 2024-03-20

**Authors:** Helder P. B. Nunes, Cristiana S. A. M. Maduro Dias, Nuno V. Álvaro, Alfredo E. S. Borba

**Affiliations:** 1Institute of Agricultural and Environmental Research and Technology (IITAA), Faculty of Agricultural and Environmental Sciences, University of the Azores, 9700-042 Angra do Heroísmo, Portugal; cristianarodrigues@gmail.com (C.S.A.M.M.D.); alfredo.es.borba@uac.pt (A.E.S.B.); 2Group of Climate, Meteorology and Global Change, Institute of Agricultural and Environmental Research and Technology (IITAA), Faculty of Agricultural and Environmental Sciences, University of the Azores, Rua Capitão João d’Ávila, 9700-042 Angra do Heroísmo, Portugal; nuno.ms.alvaro@uac.pt

**Keywords:** *Asparagopsis taxiformis*, *Asparagopsis armata*, enteric methane, ruminants, methane mitigation

## Abstract

**Simple Summary:**

This research evaluates the nutritional value and mineral content of two red algae species from the Azorean Sea, *Asparagopsis taxiformis* (native) and *Asparagopsis armata* (invasive). The aim is to assess their impact on in vitro rumen fermentation characteristics, including total gas and methane production, when added to a substrate of grass. The study finds that both algae species exhibit high protein levels (≈23.5% DM) and significant amounts of magnesium (1.15% DM) sodium (8.6% DM) and iron (2851 ppm). Furthermore, it was demonstrated that the addition of *A. taxiformis* at a concentration of 5% resulted in an 84% reduction in enteric methane production within the first 24 h, whereas *A. armata*, at the same concentration, reduced methane production by 34%. These in vitro findings suggest that *Asparagopsis* species from the Azorean Sea have potential as effective protein and mineral supplements, offering the additional benefit of reducing methane emissions from rumen fermentation.

**Abstract:**

The utilisation of seaweeds as feed supplements has been investigated for their potential to mitigate enteric methane emissions from ruminants. Enteric methane emissions are the primary source of direct greenhouse gas emissions in livestock and significantly contribute to anthropogenic methane emissions worldwide. The aim of the present study is to evaluate the nutritional role and the in vitro effect on cumulative gas and methane production of *Asparagopsis taxiformis* (native species) and *Asparagopsis armata* (invasive species), two species of red algae from the Azorean Sea, as well as the ability to reduce biogas production when incubated with single pasture (*Lolium perenne* and *Trifollium repens*) as substrate. Four levels of concentrations marine algae were used (1.25%, 2.25%, 5%, and 10% DM) and added to the substrate to evaluate ruminal fermentation using the in vitro gas production technique. The total amount of gas and methane produced by the treatment incubation was recorded during 72 h of incubation. The results indicate that both algae species under investigation contain relatively high levels of protein (22.69% and 24.23%, respectively, for *Asparagopsis taxiformis* and *Asparagopsis armata*) and significant amounts of minerals, namely magnesium (1.15% DM), sodium (8.6% DM), and iron (2851 ppm). Concerning in vitro ruminal fermentation, it was observed that *A. taxiformis* can reduce enteric methane production by approximately 86%, during the first 24 h when 5% is added. In the same period and at the same concentration, *A. armata* reduced methane production by 34%. Thus, it can be concluded that *Asparagopsis* species from the Azorean Sea have high potential as a protein and mineral supplement, in addition to enabling a reduction in methane production from rumen fermentation.

## 1. Introduction

The excellent soil and climate conditions for grass production in the area under study facilitate the rearing of dairy cattle, which is one of the pillars of the Azorean economy [1]. However, livestock production faces the challenge of reducing enteric methane emissions, since the weight of methane in the gas emissions profile of the Azores (69.1%), is higher than in mainland Portugal (52.4%) [2]. Among the various strategies studied for mitigating enteric methane produced by ruminants, the inclusion of macroalgae in the diet of ruminants stands out. Macroalgae have highly bioactive secondary compounds capable of modifying the rumen environment, interfering with the metabolism of methanogenic bacteria, resulting in lower production of enteric methane (g/day per animal) [3,4,5,6]. In addition, the incorporation of macroalgae into ruminant diets increases feed conversion efficiency by redistributing energy from the microbial methanogenesis pathway into energy pathways for the animal, such as the production of volatile fatty acids [7,8]. Among the various algae studied, those of the *Asparagopsis* genus stand out for synthesising and encapsulating methane (CH_4_) halogen analogues, such as bromoform and dibromochloromethane, which reduce the production of enteric methane in ruminants [6,9,10]. Furthermore, red macroalgae are a source of nutrients, minerals (macro and microminerals) and bioactive compounds with nutraceutical properties [11].

Currently, two red algae with potential to mitigate enteric methane can be found in the sea of the Azores archipelago, one native and one invasive: *Asparagopsis taxiformis* and *Asparagopsis armata*, respectively. These two species of red algae easily adapt to the Azorean Sea, having a wide distribution throughout the archipelago, as there are no natural predators [12]. *Asparagopsis taxiformis* is a native macroalgae of the Rhodophyta family, class *Florifeophycea*, order *Bonnemaisoniales*, and family *Bommemaisoniaceae*. *Asparagopsis taxiformis* has a heteromorphic diploid life cycle, with a haploid gametophytic stage throughout the year and a tetrasporophytic state called the Falkenbergia stage. *Asparagopsis armata* is an invasive species, present in the Azores archipelago, which was first reported by Tittley et al. [13]. This macroalgae also has a diplohaplontic heteromorphic life cycle. In this cycle, the free-living, filamentous diploid sporophyte (“Falkenbergia” stage) produces haploid spores through meiosis. These spores develop into erect, typically benthic, male, or female haploid gametophytes that generate male or female gametes. Fertilisation takes place in the female reproductive structures, resulting in a diploid zygote that matures into a multicellular carposporophyte, remaining reliant on the female thallus. It is also present in the Canary Islands [14] and Madeira [15]. Like *Asparagopsis taxiformis*, it also has a tetrasporophytic state and is confused in the ocean with the tetrasporophytic phase of *Asparagopsis armata*, making underwater identification extremely difficult.

Coupled with the imperative to mitigate enteric methane emissions from livestock farms, this leads us to believe that the inclusion of algae in cattle feed will be a promising path to achieve economic and environmental sustainability in the Azores archipelago.

The prevalence of the algae *Asparagopsis taxiformis* and *Asparagopsis armata* in the Azorean Sea is particularly concerning given the latter’s invasive nature, which poses a significant challenge when it comes to safe-guarding marine biodiversity. Coupled with the imperative to mitigate enteric methane emissions from livestock farms, this leads us to believe that the inclusion of algae in cattle feed will be a promising path for achieving economic and environmental sustainability in the Azores archipelago. Furthermore, according to Kinley et al. [16], the chemical composition of the algae varies depending on the site of collection and the season of the year. The action of algal bioactive compounds on methanogenesis depends on the type of feed supplied to ruminants. The feed also influences the microbiome of cattle [17]. Thus, in this work, we proposed local collection of these two algae, *Asparagopsis taxiformis* and *Asparagopsis armata,* and planned to characterize them nutritionally. To our knowledge, the chemical composition, and the effect of this type of seaweed on in vitro ruminal digestibility and methane production have not been determined in fresh Azorean macroalgae. We also determined the potential of these algae to mitigate the production of enteric methane in cattle through an in vitro anaerobic digestion, using only pasture as substrate, to provide the region with data in decision-making for a strategy to reduce the carbon footprint of livestock in regions where ruminants feed on pasture throughout the year.

## 2. Materials and Methods

### 2.1. Experimental Design

To determine the effect of *Asparagopsis taxiformis* and *Asparagopsis armata* on methane production, an in vitro anaerobic fermentation system was used to simulate rumen fermentation. The base substrate used in the fermenters consists exclusively of local pasture samples, with a floristic composition of 80 percent perennial ryegrass (*Lolium perenne*) and 20 percent clover pasture (*Trifolium repens*), which were dried at 65 °C in a forced ventilation oven for 72 h, ground on a 1 mm sieve in a Retsch mill (GmbH, 5657 Hann, Germany), and stored in hermetically sealed bags until use [18]. The pasture sample was used as a control (T0). Four levels of algae supplementation were tested: 1.25 percent (T1), 2.5 percent (T2), 5 percent (T3), and 10 percent (T4), i.e., part of the initial substrate was replaced with algae (considering DM), according to the treatments (Figure 1).

Samples were also evaluated to determine their chemical composition in the laboratory.

### 2.2. Seaweed Collection and Preparation

Sampling was performed at 5 m depth near the Port of São Fernando in Terceira Island (coordinates: 38°40′34′′ N, 27°03′50′′ W) in April 2022. Algae collection was conducted though skin diving when specimens were in a tetrasporophyte state, also called the “*Falkenbergia*” state. One diver would take a mesh bag and a large pair of scissors; they could then cut the specimens without pulling on them, allowing for the adhesive disk to remain in its place and providing space for the specimens to regrow. The collected items were taken to shore inside the mesh bag and accommodated in a container with sea water taken from the sampling site. The macroalgae were taken directly to the laboratory, where triage and epiphytes removal was performed before dehydration in an oven at 65 °C for 48 h.

### 2.3. Chemical Analysis and Digestibility

The chemical composition of seaweed and the pasture was determined through standard methods according to AOAC [19] for dry matter (DM, method 930.15), crude protein (CP, method 954.01), and total ash method (942.05). The neutral detergent fibre (NDF), acid detergent fibre (ADF), and acid detergent lignin (ADL) were also analysed according to Goering and Van Soest [20]. The CP content of seaweeds is calculated based on the total nitrogen content (N × 6.25). The factor of 6.25 may overestimate the crude protein content of macroalgae in general. The in vitro dry matter digestibility (DMD) and in vitro organic matter digestibility (OMD) were measured according to the method of Tilley and Terry [21], modified by Alexander and McGowan [22]. The mineral content of *Asparagopsis armata* and *Asparagopsis taxiformis* such as calcium (Ca), copper (Cu), iron (Fe), manganese (Mn), magnesium (Mg) phosphorous (P), Potassium (K), and zinc (Zn) was determined by atomic absorption spectrophotometry.

### 2.4. Rumen Fluid Collection

The rumen fluid used in the experiment was collected from the local slaughterhouse as described by Borba et al. [23]. The following conditions were observed for each experiment: Rumen fluid was collected from 5 healthy dairy cows (*Holstein* breed) with 480 ± 60 kg body weight, with an average age of 63.1 ± 13.4 months. In the last 6 weeks before slaughter, the animals were fed exclusively on pasture with a mixture of ryegrass (80%) and clovers (20%). Before slaughter, the animals were fasted for 12 h, with water available ad libitum. Rumen fluid was collected within 10 min of slaughter, filtered using cheesecloth, and preserved at 38 °C under anaerobic conditions, being delivered to the animal nutrition laboratory within 30 min of being collected. The mixture medium was prepared using the buffer solutions and the rumen fluid as described by Menke et al. [24], mixing rumen fluid with buffer solutions (reduced and mineral solutions) in a ratio of 1:2 *v/v*.

### 2.5. Gas Production and Methane Measure

To determine the total gas and methane production in ruminal fermentation, we utilised a gas fermentation system equipped with sealed bioreactors and agitation devices. We followed the protocol described by Nunes et al. [25]. In vitro tests were conducted for each alga separately, with five distinct treatments: T0—Control (no algae addition); T1—1.25%; T2—2.5%; T3—5%; and T4—10%, measuring the total gas and methane volume. Each assay was repeated three times to ensure statistical precision (*n* = 3). The ruminal fluid was diluted in a 1:2 (*v/v*) ratio in a buffer solution. Blank samples were included in each assay to control the ruminal variability. The total gas volume produced was assessed by connecting the bioreactors directly to a MilliGascounter, with a measurement range of 1 mL/h to 1 L/h and 3% precision. To measure the CH_4_ volume produced, the bioreactors were initially connected to a CO_2_ absorption unit consisting of a sealed glass flask containing 250 mL of a 3 mol/L NaOH solution (recommended by the manufacturer) to absorb CO_2_, releasing CH_4_, which was then quantified by the MilliGascounter. The bioreactors were maintained at 38 °C for 72 h. The data collected by the MilliGascounters were automatically recorded on a computer equipped with specific software (Rigamo v3.1), indicating the gas volume produced in each bioreactor.

### 2.6. Statistical Analysis

All statistical analyses were performed using the IBM SPSS Statistics for Windows, Version 27.0. Armonk, NY: IBM Corp. All data were evaluated for normality to fulfil the ANOVA assumptions. Employing the General Linear Model (GLM) with orthogonal contrasts for analysis of variance (ANOVA), we scrutinised the linear (L), quadratic (Q), and cubic (C) trends associated with four treatments. There were four treatments for each alga, each with a different concentration. The percentage refers to the quantity of the substrate/DM, T1: 1.25%, T2: 2.50%, T3: 5.00% and T4: 10.00%. One control was considered and is referred to as T0. The linear contrast assessed the systematic relationship between the concentrations of *Asparagopsis taxiformis* and *Asparagopsis armata* and the response variable. The quadratic contrast explored potential curvilinear patterns, while the cubic contrast delved into more complex, non-linear trends. The results presented express the least squares means, the standard error of means, and the contrasts’ *p*-values for the variables analysed. Significance testing at α = 0.05 revealed the unique effects of *Asparagopsis taxiformis* and *Asparagopsis armata* on the substrate, with the results considered significant if the *p*-value fell below *p*≤ 0.05 and the level of significance considered for the tendencies was 0.05 < *p* < 0.10.

## 3. Results and Discussion

### 3.1. Chemical Composition and Digestibility

The chemical composition of the two red macroalgae found in the Azorean Sea (Table 1) revealed higher levels of crude protein content in comparison to those typically encountered in pastures. The protein value for *Asparagopsis taxiformis* was 22.69% DM, in line with the value reported by Pacheco et al. [26] who, in their work, indicate a protein content of 23.76% DM for *Asparagopsis taxiformis*. However, other authors such as Brooke et al. [4] and De Bhowmick [27] also determined the crude protein (CP) *Asparagopsis taxiformis*, obtaining values ranging between 18.2 and 7.5% DM, and these algae are from regions such as Australia or Ireland. The same is the case of the macroalga *Asparagopsis armata*, which in this study showed a CP content of 24.23% DM, while Mihaile et al. [5] refers to an average CP value of 15.2% DM for algae collected in New Zealand. Parameters such as dry matter, ash, and ether extract (EE) present values similar between the two algae, and the values obtained for EE were similar to those published previously by Roque et al. [3,4,5,6].

Both seaweeds presented, in their composition, macronutrients such as calcium (Ca), phosphorus (P), magnesium (Mg), potassium (K) and sodium (Na), and trace elements of micronutrients such as copper (Cu), iron (Fe), zinc (Zn), and manganese (Mn), which are essential to the physiological metabolism of bovines [28]. Although there were differences in macronutrients such as Mg and K of approximately 28% between the two algae (Table 2), it was in the trace elements, namely Cu, Fe, Zn and Mn, that greater differences were found. The concentrations of Fe in *Asparagopsis taxiformis* were 3.8 times greater than those found in *Asparagopsis armata*. On the contrary, the Cu content found in *Asparagopsis taxiformis* (7.41 ppm) was 43% less than that found in *Asparagopsis armata* (12.99 ppm). The trace elements present in fresh seaweed may constitute an important source for the natural supplementation of trace elements for ruminants in the Azores, since there are recently published studies, such as SREAC [2] and Linhares et al. [29], indicating a generalised deficiency of trace elements in Azorean cattle, among which are manganese and copper, with the values found in both seaweeds exceeding the minimum needs required by dairy cattle [30], when considering the analysed values per gram of dry matter (g/DM). However, it is important to note that the adequacy of these trace elements to meet the minimum requirements of dairy cows depends on the quantity of seaweed consumed and the bioavailability of these trace elements in digestion, which was not evaluated in this study. The average neutral detergent fibre (NDF) content in *Asparagopsis taxiformis* was 69.81% DM, a value much higher than the one found by Roque et al. [3] and by Brooke et al. [4], who obtained 33.7 and 25.7% DM, respectively. Although the NDF content in *Asparagopsis armata* (37.88% DM) was lower than the one recorded in *Asparagopsis taxiformis*, it was higher than the one found by Mihaila et al. [5] 24.8% DM. Considering the high levels of NDF present in *Asparagopsis taxiformis*, incorporating it into bovine diets could lead to a decrease in the animals’ capacity to consume dry matter. The lignin represents the insoluble fraction of the samples, having a direct influence on their digestibility. The Acid Detergent Lignin (ADL) content present in *Asparagopsis taxiformis* was 4.02% DM, a value similar to the one found by Roque et al. [6] for this species (4.08% DM). For *Asparagopsis armata*, the same author found, in [3], a lignin content of 2.83% DM, whereas, in this study, for this same alga, we observed a higher ADL content of 10.24% DM.

### 3.2. Total Volume of Gas and Methane Produced

The introduction of seaweed in animal feed has been the subject of several studies, specifically for the mitigation of methane generated in rumen fermentation. However, previous studies [3,4,6] observed the heterogeneity at different fermentation parameters such as total gas production and CH_4_ production, among other parameters. The volume of gas production and, consequently, the amount of greenhouse gases emitted is directly influenced by ruminants’ diet, feed digestibility, and algae administration [6,7,8,9,10,11,12,13,14,15,16]. This in vitro study is the first to consider the total production of gas and CH_4_ using different percentages of red macroalgae from the Azores (Terceira Island) and using exclusively pasture-based feed (substrate), which was chemically characterised (Table 1), emphasising the fibre content. This aspect is crucial to understand due to the observed positive correlation between fibre content and enteric methane production [31]. 

The inclusion of these algae enabled a reduction in total gas production over 72 h. However, at different concentrations, *Asparagopsis taxiformis* demonstrated superior performance compared to *Asparagopsis armata* in terms of reducing total gas production (Table 3). The contrast values at 6 h of incubation of 0.199 (cubic), 0.228 (quadratic), and 0.086 (linear) highlight the significance of these trends. The most substantial reduction in total gas production was observed when 10% algae (T4) were introduced into the substrates, resulting in an average reduction of 48.9% for *Asparagopsis taxiformis* and 22.8% in *Asparagopsis armata* (Table 3). These findings were further supported by contrast analysis, revealing significant patterns in the data over the incubation period. Specifically, a linear contrast of 0.035 for *Asparagopsis armata* indicates a linear trend in the reduction of total gas production, while a quadratic contrast of 0.049 suggests a curvilinear effect, which is particularly evident in *Asparagopsis taxiformis* during 72 h incubation period. The most significant (*p* < 0.05) reduction in total gas production occurred within the first 24 h of fermentation, highlighting a notable difference between the volume of gas produced in the treatment groups compared to the control group. For *Asparagopsis taxiformis,* the greatest reduction in total gas production was observed at 12 h, progressively decreasing throughout the 72 h duration of the experiment (quadratic effect = 0.049). Even at 72 h, the total gas production remained significantly lower (*p* < 0.05) than the gas production in the control group. Our results are in line with those published by Kinley et al. [16], who also observed a significant decrease in total gas production when adding equal values of 2% organic matter or higher of *Asparagopsis taxiformis*. In *Asparagopsis armata*, the peak reduction in gas occurred at 24 h, with statistical differences (*p* < 0.05) observed in gas production when comparing the different concentrations with the control group. A study conducted by Mihaila et al. [5], in which 5% organic matter from *Asparagopsis armata* was added to a lyophilised perennial ryegrass substrate, reported significant differences in total gas production at 24 h. Our study extended to 72 h, revealing that after 48 h of fermentation, there was a significant increase (*p* < 0.05) of over 50% in total gas production at concentrations 1.25%, 2.5%, and 5%. At 48 h, reduction rates of less than 6% were recorded, with no significant differences between the control and concentrations below 5%. These results indicate that the addition of *Asparagopsis armata* influenced the general microbiological metabolism of the rumen in the first 24 h, with a reduction in gas production. However, beyond this period, there was a re-establishment of the microbial population in the rumen capable of maintaining homeostasis, suggesting the preservation of a healthy ruminal flora. This preservation resulted in gas production similar to that produced in the control group, especially at the lower concentrations.

Regarding the production of methane, we can observe from Table 4 that the addition of *Asparagopsis taxiformis* to the substrate based in grass enabled a significant reduction in methane production throughout the 72 h of fermentation, regardless of the DM concentration used. The highest methane reduction (88.9%) was observed at 12 h after including 5% of *Asparagopsis taxiformis* in the substrate, in relation to the control group. The statistical analysis revealed a significant linear effect (*p* < 0.001) and a quadratic effect (*p* = 0.067) over the incubation time, indicating a consistent and potentially dose-dependent impact on methane reduction. In case of *Asparagopsis armata*, statistically significant (*p* < 0.05) reductions in methane production were observed only when 5% and 10% algae were added to the feed during the 72 h period. Concentrations lower than 5% algae showed, on average, higher methane production at 6 and 72 h compared to the control group. This suggests a longer lag phase for these lower concentrations to influence rumen microorganisms, with a deceleration of their activity before 72 h of fermentation. The statistical analysis confirmed a significant linear effect (*p* = 0.035) and a quadratic effect (*p* = 0.568) for *Asparagopsis armata*, indicating a nuanced relationship between concentration and methane reduction. The addition of 5% *Asparagopsis taxiformis* reduced methane production at 24 h by approximately 86% (*p* < 0.001). This value falls within the range reported by Roque et al. [3] and by Broke et al. [4], who obtained methane reductions of 95% and 78%, respectively. 

For *Asparagopsis armata*, during the same timeframe (72 h) and at the same identical inclusion rate (5%), methane reduction amounted to approximately 34% compared to the control group. However, a not statistically significant (*p* > 0.05) decrease in CH_4_ production or total gas production was observed when including this alga at 1.25% DM. However, previous in vivo study in cattle using *Asparagopsis armata*, showed that methane production decreased by 42.7% at a level of 1% dry matter inclusion in the diet [3], although it should be noted that the substrate composition of this trial was different from the one used. As highlighted by Lean et al. [32] in their meta-analysis, in which they evaluated the use of the genus *Asparagopsis* in the inclusion of ruminant diets, there is a marked heterogeneity in the results of methane reductions, with the action of algae depending on multiple factors, including the type of ruminant diet. In the specific case of the Azores archipelago, more than 95% of the 278,000 cattle existing in the region [33] feed directly from pastures, emitting around 21,500 tonnes of carbon of enteric origin annually [2]. The algae considered in the present study are rich in protein and trace elements, specifically of manganese and copper, these marine macroalgae could contribute to mitigating enteric methane production.

## 4. Conclusions

In conclusion, the results obtained in this study corroborate the potential of using fresh red macroalgae from the Azores Sea as a promising way to mitigate methane emissions in ruminant livestock while offering a sustainable protein source rich in essential trace elements such as manganese and copper. The inclusion of 5% *Asparagopsis taxiformis* in the substrate led to a significant reduction in methane production of around 86% within 24 h, highlighting its effectiveness in containing greenhouse gas emissions. However, it is essential to carry out further research to reinforce the results obtained and verify the practical viability and effectiveness of implementing these algae-based supplements in diets for ruminants fed on pasture. Furthermore, exploring potential refinement techniques for *Asparagopsis armata* to improve its digestibility could further optimise its use in animal feed formulations, since the NDF content present in macroalgae can play a crucial role in modulating digestibility and the balance of the rumen microbiota. In general, harnessing the potential of these marine red macroalgae from the Azores Sea not only presents a viable strategy for reducing the environmental footprint of ruminant agriculture, but can also be a source of protein and minerals, aligning with the broader objectives of promoting sustainable agricultural practices and fostering a circular economy.

## Figures and Tables

**Figure 1 animals-14-00967-f001:**
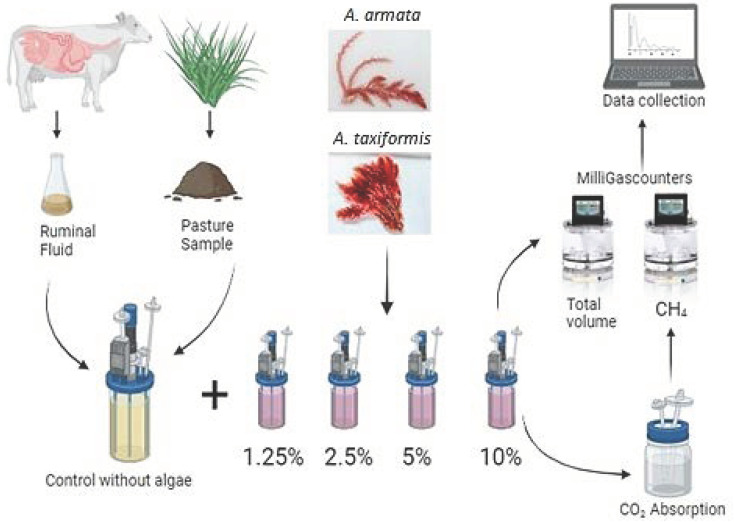
Schematic of the experimental design used in this experiment. Samples of rumen fluid were gathered from animals slaughtered at a local abattoir, while pasture samples were collected and subjected to analysis to serve as substrate. Ruminal liquid with substrate was added to each fermenter and different amounts of algae (1.25%; 2.5%; 5%; and 10%), replacing part of the initial substrate. Each fermenter was connected to a flow meter, measuring the volume of gas produced.

**Table 1 animals-14-00967-t001:** Nutritional composition of *Asparagopsis taxiformis, Asparagopsis armata*, substrate, and the substrate used in fermentation.

Parameters	DM (%)	CP (%DM)	NDF (%DM)	ADF (%DM)	ADL (%DM)	EE (%DM)	Ash (%DM)	DMD (%)	OMD (%)
*A. taxiformis*	6.55 ± 1.25	22.69 ± 5.96	69.81 ± 5.12	15.05 ± 2.08	4.01 ± 0.96	0.53 ± 0.02	37.5 ± 1.56	68.45 ± 10.00	55.30 ± 3.79
*A. armata*	7.68 ± 1.07	24.23 ± 4.26	37.88 ± 7.23	12.04 ± 1.87	10.24 ± 1.22	0.38 ± 0.03	36.7 ± 1.98	40.61 ± 8.04	34.93 ± 4.10
Basal Diet	10.35 ± 2.49	19.79 ± 3.99	64.14 ± 4.01	33.9 ± 4.40	3.34 ± 0.43	1.09 ± 0.02	14.56 ± 0.87	76.03 ± 6.54	72.41 ± 3.21

Values represent the mean ± standard deviation. DM—dry matter; CP—crude protein; NDF—neutral detergent fibre; ADF—acid detergent fibre; ADL—acid detergent lignin; EE—ether extract; Ash—Crude ash; DMD—dry matter digestibility; OMD—organic matter digestibility.

**Table 2 animals-14-00967-t002:** Macro and microminerals composition of *Asparagopsis taxiformis, Asparagopsis armata,* and substrate.

Parameters	Ca	P	Mg	K	Na	Cu	Fe	Zn	Mn
	%DM	%DM	%DM	%DM	%DM	ppm	Ppm	ppm	Ppm
*A. taxiformis*	3.7 ± 0.07	0.20 ± 0.06	0.96 ± 0.09	2.01 ± 0.49	8.41 ± 0.62	7.00 ± 0.68	4524.67 ± 342.65	38 ± 3.56	110.7 ± 25.67
*A. armata*	3.9 ± 0.05	0.21 ± 0.02	1.35 ± 0.08	1.45 ± 0.25	8.72 ± 0.58	12.90 ± 0.24	1178 ± 225.87	65.7 ± 12.2	61 ± 12.95
Substrate	0.42 ± 0.05	0.18 ± 0.04	0.25 ± 0.02	2.17 ± 0.71	0.31 ± 0.09	8.86 ± 0.54	587.62 ± 120.44	47.21 ± 9.99	99.4 ± 19.81

Values represent the mean ± standard deviation. DM—dry matter; Ca—calcium; P—phosphorus; Mg—magnesium; K—potassium; Na –sodium; Cu—copper; Fe—iron; Zn—zinc; Mn—manganese.

**Table 3 animals-14-00967-t003:** Effects of different treatments of *Asparagopsis armata* and *Asparagopsis taxiformis* extracts on in vitro cumulative gas production by mixed rumen anaerobic fermentation for 72 h.

Incubation Time (h)	Treatment					SEM	Contrast		
	T0	T1	T2	T3	T4		Linear	Quadratic	Cubic
	Total Gas Production (mL/g DM of Substrate)								
*Asparagopsis armata*									
6	28.01	24.64	25.10	24.77	25.18	0.64	0.086	0.228	0.199
12	66.63	63.01	59.74	59.98	44.56	0.29	0.315	0.888	0.764
24	125.86	115.18	97.33	93.24	75.52	0.32	0.101	0.035	0.085
48	157.70	156.81	153.85	149.00	128.40	0.38	0.047	0.024	0.032
72	175.32	168.94	169.34	159.50	153.85	0.68	0.035	0.044	0.031
*Asparagopsis taxiformis*									
6	33.65	29.32	23.05	21.54	12.72	0.64	0.029	0.285	0.217
12	70.29	59.41	46.05	41.14	23.03	0.29	<0.001	0.423	0.051
24	136.85	118.41	95.78	89.52	73.25	0.32	<0.001	0.224	0.057
48	173.70	151.45	128.41	113.94	105.31	0.38	<0.001	0.247	0.046
72	180.92	169.96	154.92	141.55	127.76	0.68	<0.001	0.049	<0.001

SEM—standard error of the mean, *n* = 3; DM—dry matter; h—hour; T0—control; T1—treatment with 1.25% algae; T2—treatment with 2.5% algae; T3—treatment with 5% algae; T4—treatment with 10% algae.

**Table 4 animals-14-00967-t004:** Effects of different treatment of *Asparagopsis armata* and *Asparagopsis taxiformis* Rhus succedanea extracts on in vitro cumulative methane production by mixed rumen anaerobic fermentation.

Incubation Time (h)	Treatment					SEM	Contrast		
	T0	T1	T2	T3	T4		Linear	Quadratic	Cubic
	Methane Production (mL/g DM of Substrate)								
*Asparagopsis armata*									
6	7.24	8.20	7.41	3.65	3.60	0.64	0.658	0.682	0.407
12	14.52	14.08	14.42	10.84	6.94	0.29	0.042	0.423	0.687
24	27.60	26.69	22.83	18.22	16.38	0.32	0.032	0.224	0.312
48	32.24	32.20	30.77	25.62	22.26	0.38	0.047	0.247	0.225
72	34.42	34.62	35.10	28.46	25.97	0.68	0.035	0.568	0.722
*Asparagopsis taxiformis*									
6	7.81	2.68	2.02	1.27	1.34	0.05	0.044	0.487	0.684
12	12.08	7.62	4.69	1.34	2.71	0.08	0.039	0.325	0.551
24	24.04	14.73	11.59	3.13	5.77	0.12	0.027	0.158	0.256
48	28.33	19.44	14.75	6.42	7.88	0.26	<0.001	0.067	0.088
72	29.28	20.92	16.54	7.55	9.28	0.42	<0.001	0.048	0.037

SEM—standard error of the mean, *n* = 3; DM—dry matter; h—hour; T0—control; T1—treatment with 1.25% algae; T2—treatment with 2.5% algae; T3—treatment with 5% algae; T4—treatment with 10% algae.

## Data Availability

The data presented in this study are available on request from the corresponding author.

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
