# Peer review of "Evaluation of Two Species of Macroalgae from Azores Sea as Potential Reducers of Ruminal Methane Production: In Vitro Ruminal Assay"

_animals, 2024, doi:10.3390/ani14060967_

Round 1
Reviewer 1 Report
Comments and Suggestions for Authors
As we known, there was a in vivo study confirmed that enteric CH4 could be virtually eliminated while using Asparagopsis taxiformis as a feed ingredient in the high grain TMR of feedlot beef cattle (Kinley et al., 2020). Generally, in vitro experiments are done before in vivo experiments, given the maneuverability of the experiments, and there is already evidence of in vivo experiments, so what is the significance of doing in vitro experiments again? Please elaborate on the purpose of your study.
Line 17 “un” should be “an”.
Line 29-30 “To each experimental unit, 800 ml of rumen inoculum and 6g of DM of basal diet were added.” should be deleted.
Line 34 You can add evaluated crude protein value after “have high levels of protein”.
Line 37 “by 34% DM”, please delete “DM”.
Line 79 “by [13]” should be “by Tittley et al. [13]”. There are several similar errors in the article, including line 133 “according to [18]”, line 136 “according to the method of [19]” and so on, so please proofread carefully and correct them.
Line 97 “the nutritional value these seaweeds” should be “the nutritional value of these seaweeds”.
Line 98 Please delete “propose to”.
Line 115 “5m” should be “5 m”. line 157 “800ml” should be “800 ml”, “6g” should be “6 g”. There are several similar errors in the article, so please proofread carefully and correct them.
Line 162 “Dr. Ing. Ritter Apparatebau GmbH & Co. KG”, the information about MilliGascounter was incorrect. line 180-181 “IBM®, SPSS Statistics Version 27 program (SPSS Inc., Chicago, IL, USA)” the information about SPSS was incorrect. Please check the format, refer to the articles published.
Line 164-166 and line 186, “CO2” should be “CO2”. Line 187 “H2” should be “H2”
Line 164 “3M” should be “3 mol/L”.
Line 169-171 “(Rigamo v3.1, 2012 Dr. Ing. Ritter Apparatebau GmbH & Co. kg)”, the information about the software was incorrect, and “specific software” should be “Rigamo v3.1 (the information of Rigamo v3.1)”.
Line 206 “like” should be “consist with”.
Line 208 “Km” should be “km”.
Line 207-209 and line 240 “A. taxiformis” should be “A. taxiformis”.
Line 211-212 Please use the full name of “PB”
Line 223 “essential to the physiological metabolism of bovines”, please add the references.
Line 256 “ To our knowledge” should be “To our knowledge”.
Line 259-260 I can’t understand the meaning of “highlighting the fibre content, which is essential to know since there is a positive correlation between fibre content and enteric methane production”, please further explain it.
Line 268 “at the different concentrations” should be “at different concentrations”.
Line 283 and line 286 Please use the full name of “MO”
Line 334 “Asparagopsis” should be “Asparagopsis” as genus.
Line 338-340 You multiply 21,500 by 86% to get 7,300, and 21,500 by 34% to get 18,500, but that's the wrong way to estimate it. This is because 86% and 34% are the maximum reduction rates, but they cannot be achieved at other time points, as shown in Figure 4..
Line 342-343 I can’t understand the meaning of “transposing the results obtained in vitro into in vivo”, please further explain it.
Author Response
Dear reviewer,
Thank you for your contribution to improving the manuscript. His suggestions were very helpful in overcoming the weaknesses of this manuscript.
Attached are the changes, point by point, that were made and the answers to your questions.
We hope this has been clarified and we are available to answer any further questions that may arise.

Reviewer 2 Report
Comments and Suggestions for Authors
Dear Authors,
the abstract needs more clarity regarding the results. Present the results obtained more completely. Review the amount of written methodology and introduction.
In introduction, it´s necessary to reduce first pharagraph (line 46-69).
In methodology, item 2.4. Rumen Fluid Collection, which animal welfare standard was applied?
Which animal welfare standard was applied? Are animals fistulated? Was there a need to perform surgery to open the fistula in these animals? If yes, please describe how it was carried out, in accordance with animal welfare standards. (Line 143-152). Also consider collecting rumen fluid.
Item results, in first paraghaph is review. Please, remove and follow with results your study.
Line 249-252: What do the authors recommend with this result obtained?
Line 259: Which studies? Cite references.
Line:264-266: This phrase has already been quoted.
The last pagraph is too long.
Line 350: 'Transposing these in vitro results into in vivo scenarios, 350 the addition of 2.5% Asparagopsis taxiformis to the diet of Azorean cattle would allow for 351 a 50% reduction in the enteric methane emissions., transposing the results obtained in vitro 352 into in vivo'. How can the authors claim this without conducting the experiment in vivo? Please review this statement.
In conclusion, and the value de NDF? Wich conclusion?
Author Response
Dear reviewer,
Thank you for your contribution to improving the manuscript. Your suggestions were very useful in overcoming the weaknesses of this manuscript.
Attached, I will now indicate point by point what has been changed and answer your questions when raised.
The authors' responses are identified by "AR".
The alterations in manuscript were marked in red.

Reviewer 3 Report
Comments and Suggestions for Authors
This article focuses on a topic of high relevance and actuality. The implementation of functional additives in ruminants, improving the sustainability of animal production could enhance the supply chain. However, in the present article there are several implementations to be carried out.
In all the text there are multiple formatting and punctuation errors, i.e. line 16, 17, 25, 35, 36, 37, 39 etc.
An English revision needs to be performed.
Change the abstract introduction, the algae are able to increase the nutritional value of feed but have not the ability to transform alternative feeds. However, ruminants have the capacity of transforming low nutritional value forages into high nutritional value products such as milk or meat.
The material and method section should be implemented, no information about the Approval of an Animal Care Protocol (approvation N°) are reported.
In addition, some important information regarding animals are missing. For example, age of animals, body weight, the 5 animals selected were sufficient for the study protocol? The information regarding animals’ diet is not clearly specify. When the last feeding was done? How much was the dry matter intake?
Some information regarding the substrate adopted are missing. Sometimes authors are use the terms substrate referred to the diets, sometimes is used pasture, this makes the reader confused.
The in vitro protocol is not fully clear, for example seems confused the explanation of the protocol with regard to the substrate adopted and for algae inoculum. I am wondering if the method adopted is suitable for methane quantification or can be adopted for estimate the quantity of this gas by transforming CO2.
All the Table and the Figure in the text should be standardised and formatted.
The conclusion should be rewritten.
Lines 17-18: please modify to “A. taxiformis at 5% concentration showed a reduction”.
Lines 19-20: please add “These in vitro findings”
Lines 23-25: The algae can increase the nutritional value of animal’s feed but have not the ability to transform alternative feeds. Replace the abstract introduction with the following sentence:
“The implementation of algae in animals’ diet have a beneficial impact in multiple aspects, first algae are known for their composition, rich in polyphenols, antioxidant activity, minerals, and vitamins, characterized by high nutritional level. Asparagopsis taxiformis and Asparagopsis armata are the two algae involved in the study and are investigated to evaluate the mitigation the production of enteric methane. “
Lines 25-29: The sentence “The aims of the present study are to determine the nutritional value and minerals content of two species of red algae from the Azorean Sea: one native (Asparagopsis taxiformis) and one invasive (Asparagopsis armata). The study also aims to evaluate the effects of their addition to the in vitro rumen fermentation characteristics through total gas and methane production, …” should be replaced by “The aim of the present study is to evaluate the nutritional role and the in vitro effect on rumen fermentation characteristics and methane production of Asparagopsis taxiformis (native species) and Asparagopsis armata (invasive species), two species of red algae from the Azorean Sea, ...”
Line 30: How many experimental units were used?
Line 32: Add the percentage of each concentration.
Line 34: The sentence “every hour after incubation for up to 72 hours” should be replaced with “once an hour, for the next 72 hours of incubation”.
Line 35-36: Delete “and elevated levels of” and replace with a “,”. Suggest to add data about protein and minerals.
Line 47: However, the regions “of Azores” …
Line 48: “it is necessary” should be replace with “this makes it necessary”.
Line 48: “for animal feed” should be replace with “ensuring a proper nutrition of animals”.
Lines 49-51: “reducing food imports, increasing the capacity to transform alternative feeds of low commercial value into foods with a 50 high nutritional level, and mitigating the production of enteric methane” should be replaced with “reduce food and feed imports, increase the nutritional value of animals meal, and mitigate the production of enteric methane.”
Line 54: the word “including” should be deleted.
Lines 55-57: Better specify the differences between gas emission profile (in numerical terms) in Azores and in mainland Portugal area.
Line 58: the word “biological” should be deleted. Algae is recognised as novel food by EFSA, and can be added to feed as additive named “aquatic products of plant origin”. The use of “biological” seems not correct.
Line 62-65: Better explain the characteristics of macroalgae used. Several are the beneficial aspects of algae, not only the methanogenic action. The minerals characterisation is not mentioned in the introduction which should be added.
Line 67: Replace “. Due to” with “for”
Line 68: synthetize.
Line 74: The explanation of Asparagopsis taxiformis is correct but is not written the Asparagopsis armata characteristics.
Line 75-76: “the species have a cosmopolitan distibution”
Line 84-85: The sentence has already been repeated earlier. It should be delated.
Line 91-92: Rewrite the sentence.
Line 95: replace the word “subjected” with “fed”.
Line 95-96: Replace “, as the microbiome of cattle varies depending on what they are fed.” with “. The feed also influences the microbiome of cattle.”
Line 99: Put scientific names in italics.
Lines 109-110: delete the sentence “in a factorial design 2x4+1” or clarify the design adopted.
Line 107-108: Replace the sentence from “To determine the effect of two macroalgae species originating in the Azorean Sea, Asparagopsis taxiformis and Asparagopsis armata, on methane production,” to “To determine the effect of Asparagopsis taxiformis and Asparagopsis armata on methane production,”.
Line 113: Correct the sentence from “perennial ryegrass (Lolium perenne) pasture” to “perennial ryegrass pasture (Lolium perenne)”.
Line 114: replace the word “taken” to “also evaluated”.
Line 112-115: the sentence is not completely clear, some information about substrate is missing, where the substrate come from? How the samples of the substrate were collected? How the authors have calculated that the substrate or pasture was 80% and 20% for the grass reported?
Line 116, Figure 1: The Figure 1 does not match with the text, the substrate is added in ruminal fluid or is ingested by the animals?
Line 117: Better explain the figure 1 in the caption.
Line 142: Add the point at the end of the sentence.
Line 135: “Were also analyzed”
Line 145: replace “For” with “for”.
Line 146: Better explain the choice of 5 animals. How and how long were the animals monitored? How were the animals handled at the slaughterhouse?
Line 147: Why was the rumen fluid collected 10 minutes before slaughter and not after slaughter?
Line 150: Better explain the method described by [22].
Line 160: Better describe the protocol described by [23]. How is the protocol adapted to the use of algae?
Line 161: 6g of algae DM sample?
Line 163: replace “consisted only of pasture (substrate)” with “is composed by substrate without algae addiction”.
Lines 163-164: How were incorporated the substrate?
Line 165: “ml/h” should be replaced to “mL/h”.
Line 174: The point should be removed.
Line 181-183: the sentence “The treatments consisting in add different concentrations of seaweeds to substrate used (T0- Control, T1 –1.25% sample, T2 –2.5% sample, T3 – 5% sample, T4 - 10% sample)” should be replaced with “The treatments are four for each alga at different concentration. The percentage is referred to the quantity of substrate/DM, T1: 1.25%, T2: 2.50%, T3: 5.00% and T4: 10.00%. One control is considered and is called T0.”.
Lines 190-191: “P” should be replaced with “p”.
Line 207: The space should be removed.
Lines 208-209, Table 1: The chemicals characterisation of each diet at different concentrations should be performed and added to the Table 1. The SD and the p-value should be added to the Table 1. Add a new row entitled IU and separate the measurement unit.
Table 1: The protein content in substrate is 19.79, a high value compared to the values suggested by NRC 2021. What is the reason? I suggest to add nutritional value of the diets plus algae in order to verify the differences among the diets. The addition of algae could increase the level of nutrients in the diet?
Lines 212-255: The statistic part should be added, show any significant results (i.e. p-value).
Line 215: “like” should be replaced with “in line with”.
Line 218: “CP” should be replaced with “Crude Protein (CP)”.
Line 221: the definition of acronym “PB” should be defined.
Line 225: “micro mineral” should be replaced with “microminerals”.
Lines 225-227, Table 2: Like Table 1, the minerals characterisation of each diet at different concentrations should be performed and added to the Table 1. The SD and the p-value should be added to the Table 2. Add also a new row entitled IU and separate the measurement unit.
Line 243: the word “us” should be deleted.
Line 258-259: the sentence “However, we have observed in the studies already published that there is heterogeneity in the results of different fermentation parameters such as total gas production, CH4 production, among others.” should be replaced with “However, in previous studies is observed the heterogeneity at different fermentation parameters such as total gas production, CH4 production, among others.”.
Line 261-262: The sentence “In addition to the chemical composition of the algae themselves, as mentioned above, other factors such as the ruminant diet and feed digestibility, directly influence the volume of gases produced and consequently the amount of greenhouse gases emitted [6-16].” should be replaced with “The volume of gas production and consequently the amount of greenhouse gases emitted is directly influenced by ruminants diet, feed digestibility and algae administration [6-16].”
Line 264-265: The sentence “To our knowledge, this is the first study to determine, in vitro, the total production of gas and CH4, using different percentages of fresh red macroalgae from the Azores” should be replaced with “This in vitro study is the first that consider the total production of gas and CH4, using different percentages of red macroalgae from the Azores”.
Line 269, Table 3: Specify in the table what is the gas evaluated. LSMeans should be added to the Table 3. Remove the “h” near all the number, it has already been said under the incubation time. The acronyms should be added to the Table 3 and the whole word should be written in the line 271. “ml/g” should be replaced with “mL/g”. The meaning of T0, T1, T2, T3, and T4 should be added line 270.
Line 273: The words “our data indicate that, on average,” should be removed.
Line 276: Remove the point.
Line 277: What is meant by “when 10% algae were introduced in the DM”?
Line 283: The common should be removed and the total gas should be specified.
Line 285: If the differences between the control group is significant in all the treatment groups the p-value should be written.
Line 287: “:” should be replaced with “=”.
Lines 290, 294: p<0.05 should be written in italics.
Lines 292, 296: Define the acronyms “MO”.
Line 295: Remove the “p”.
Line 297: The point should be removed.
Line 299: Add the percentage to 1.25 and 2.5.
Line 303: replace “24h” with “24 hours”.
Line 304: replace "reestablishment” with “re-establishment”.
Line 308-310, Table 4: Like Table 3, LSMeans should be added to the Table 4. Remove the “h” near all the number, it has already been said under the incubation time. The acronyms should be added to the Table 4 and the whole word should be written under the table. “ml/g” should be replaced with “mL/g”. The meaning of T0, T1, T2, T3, and T4 should be added in the title. Under the table add also all the acronyms visible in Table 4.
Line 314: Is the reduction of 88.9% referred to the control group?
Line 319-320-332: 10% DM of substrate? 5% DM of substrate? 1.25% DM of substrate?
Line 325: Remove the comma.
Line 327: Put scientific names in italics.
Line 332: The words “in an” should be replaced with “previous”.
Line 335: The words “by us” should be removed.
Lines 335-342: These sentences are out of topic,
Lines 353-354: The sentence “This underscores the significant potential of Asparagopsis taxiformis and Asparagopsis armata for inclusion in cattle feed in the Azores.” should be removed.
Lines 354-357: The sentence “Apart from being alternative sources of protein and trace elements, specifically of manganese and copper, these marine macroalgae from the Azorean Sea, can contribute to mitigating enteric methane production.” Should be replaced with “The algae considered in the present study are rich in protein and trace elements, specifically of manganese and copper, these marine macroalgae could contribute to mitigating enteric methane production.”
Comments on the Quality of English LanguageI suggest to revise the manuscript.
Author Response
Dear reviewer,
We greatly appreciate the revisions you made; they greatly improved the quality of the manuscript. The thoroughness of detecting errors allows this work to be valued, and all suggestions are accepted.
Let us hope that the changes have met your expectations.
The authors' responses are identified by "AR" to each of the questions/suggestions made.
All changes made were marked in red in the manuscript.

Round 2
Reviewer 1 Report
Comments and Suggestions for Authors
No further comments.
Author Response
The reviewer had no further comments to make, we thank you for your collaboration in improving the manuscript.
Reviewer 3 Report
Comments and Suggestions for Authors
Authors did not report the new lines numeration after revision in the comments, I suggest to report the number of lines for better comprehension.
Line 25-26: “the mitigation and the production”, please review English
Line 27: In vitro in italic.
Line 48: “produce cattle”, please correct to “rearing cattle”
Line 113: in the figure the first + should be deleted and the algae names should be in italics.
Previous comment: “In addition, some important information regarding animals is missing. For example, age of animals, body weight, the 5 animals selected were sufficient for the study protocol? The information regarding animals’ diet is not clearly specify. When the last feeding was done? How much was the dry matter intake? “
Answer: “AR: More detailed information has been introduced. Regarding the animals, for each test ruminal fluid was collected from 5 animals, that is, in total it was collected from 15 animals. “
New comment: Line 146-149: authors only reported the body weight of animals, I highly suggest to report the other information about age and if the number of animals where enough to perform the study. I do not understand why the authors reported in the comments that in total was collected from 15 animals, please better explain
Line 150; line 260; line 342: Remove doble dot.
Line 199-200: please review English
Line 203-204: “This protein value was obtained for the archipelago of Madeira, situated approximately 965 Km from the Azores”. The protein value? Please review the sentence
Line 208: “PB content”, please correct to “CP content”
Line 199- 212 and 225-249: Authors reported chemical and macronutrients analysis of algae, do not seem that a statistical analysis was performed; if so, it is not correct use in the text expressions as “not present significant variations” (Line 211) or “allow to observe differences between” (line 238) or “a significantly higher ADL content “(Line 249). Why authors did not report statistical analysis?
Previous comment: Table 1: The protein content in substrate is 19.79, a high value compared to the values suggested by NRC 2021. What is the reason? I suggest to add nutritional value of the diets plus algae in order to verify the differences among the diets. The addition of algae could increase the level of nutrients in the diet?
Answer: AR: The protein content of the substrate is that present in the Azores pasture, as a rule, in spring we can reach 22-24% protein. As the substrate used is exclusively pasture, we cannot change the balance of the protein content according to the NRC values.
As in this test the added value is very low, the change in chemical composition is not significant, hence it does not change the level of nutrients in the diet.
Previous comment: Line 277: What is meant by “when 10% algae were introduced in the DM”?
Answer: AR: DM was replaced with substrates.
New comment:
Authors reported in Line104: “Four supplementation levels of each algae were tested (as a percentage of substrate DM) as follows: 1.25% (T1), 2.5% (T2), 5% (T3), and 10% (T4)”; and in Lines 106: “The substrate used in the fermenters consists exclusively of local grassland, with a floristic composition of 80 107 % perennial ryegrass (Lolium perenne) and 20 % clover grassland (Trifolium repens)”. I highly suggest to better explain the algae utilization, because as reported in the text seems that the different percentages of algae were added in a different level on a “basal diet” named the pasture/substrate (this is still confused in the text even if corrections where done).
Then, authors reported in the comment “DM was replaced with substrates” which means that the dry matter of the “basal diet” named the pasture/substrate was REPLACED with algae in different percentage.
Make the article clear is a pillar in order better communicate a valuable research topic, in light of my last comment about “algae replaced part of the DM of the pasture” I suggest to underline this aspect in the aim of the study is not even reported (Line 91-99)
I disagree with the answer reported by authors about “As in this test the added value is very low, the change in chemical composition is not significant, hence it does not change the level of nutrients in the diet “. Authors did not perform chemical analysis of basal diet” named the pasture/substrate plus algae, if you replace 10% of DM with algae is recommended to test if there are change in nutrients. Then, authors reported that the “change in chemical composition is not significant” how authors can state if there are no evidence reported about?
In all the text, in particular in Table 1 and 2 the spaces between the plus-minus sign should be added, i.e. from “6.55±1.25” to “6.55 ± 1.25”.
Comments on the Quality of English LanguageModerate editing of English language required
Author Response
Once again, we thank you for your contribution to improving the manuscript.
Attached we send the answers to your questions.
